# The Alien Plants That Threaten South Africa's Mountain Ecosystems

Kim Canavan [1,*], Susan Canavan [2], Vincent Ralph Clark [3], Onalenna Gwate [3], David Mark Richardson [4], Guy Frederick Sutton [1] and Grant Douglas Martin [1,5]

[1] Centre for Biological Control, Department of Zoology and Entomology, Rhodes University, Makhanda 6140, South Africa; g.sutton@ru.ac.za (G.F.S.); g.martin@ru.ac.za (G.D.M.)
[2] Department of Invasion Ecology, Institute of Botany, Czech Academy of Sciences, 25243 Průhonice, Czech Republic; Susan.Canavan@ibot.cas.cz
[3] Afromontane Research Unit and Department of Geography, University of the Free State, Phuthaditjhaba 9866, South Africa; clarkvr@ufs.ac.za (V.R.C.); 2019868948@ufs4life.ac.za (O.G.)
[4] Centre for Invasion Biology, Department of Botany and Zoology, Stellenbosch University, Stellenbosch 7600, South Africa; rich@sun.ac.za
[5] Afromontane Research Unit and Department Zoology and Entomology, University of the Free State, Phuthaditjhaba 9866, South Africa
* Correspondence: K.Canavan@ru.ac.za

**Abstract:** The six major mountain ranges in South Africa support critically important ecosystem services—notably water production—and are rich in biodiversity and endemism. These mountains are threatened by detrimental land uses, unsustainable use of natural resources, climate change, and invasive alien plants. Invasive alien plants pose substantial and rapidly increasing problems in mountainous areas worldwide. However, little is known about the extent of plant invasions in the mountains of South Africa. This study assessed the status of alien plants in South African mountains by determining sampling efforts, species compositions and abundances across the six ranges in lower-and higher-elevation areas. Species occurrence records were obtained from three databases that used various approaches (roadside surveys, citizen science observations, focused botanical surveys). Most mountain ranges were found to be undersampled, and species composition assessments were only possible for two ranges. The majority of abundant alien plants in both the lower- and higher-elevation areas were species with broad ecological tolerances and characterised by long distance seed dispersal. These prevalent species were mostly woody plants—particularly tree species in the genera *Acacia*, *Pinus*, and *Prosopis*—that are contributing to the trend of woody plant encroachment across South African mountains. We suggest improved mountain-specific surveys to create a database which could be used to develop management strategies appropriate for each mountain range.

**Keywords:** alien species; biological invasions; citizen science; elevation; species abundance; tree invasions; woody plant encroachment

## 1. Introduction

The mountains of South Africa support critically important ecosystem services—notably water production [1,2]. Through orographic influence, mountains trap moisture, providing surface and groundwater that is essential for downstream agriculture and the persistence of major urban and industrial centres [3]. The topographic complexity of South African mountains and their distribution across a strong climatic gradient in the region has resulted in diverse ecosystems and high local endemism [4–6]. This biodiversity is critical in supporting the livelihoods of rural local communities that are often poorer and more marginalised [7], and are therefore directly reliant on natural resources, such as for agriculture and traditional medicine [8,9]. Yet montane areas are under threat from

detrimental land-uses, unsustainable use of natural resources, climate change, and invasive alien plants [10–13]. The establishment of alien plants in particular poses a substantial and continuously increasing problem in driving ecosystem changes and biodiversity loss in mountains [14]. This threat has not been adequately acknowledged in South Africa, and most mountain ranges are understudied [15]. Canavan et al. [13] highlighted that mountains remain areas of low priority for alien plant management and are generally considered to be largely resilient to invasion.

In the last two decades, research focused on mountain invasions has expanded worldwide [14,16,17]. Global reviews reveal a pattern of declining alien plant richness with increasing elevation [17–20]. This pattern has been attributed to a number of different mechanisms, including the introduction of alien species predominantly to low elevations, coupled with environmental filtering as species spread towards higher elevations [21], and limited propagule pressure [22]. Although mountains generally support fewer invasive alien plants than lowlands, there is clear evidence that alien plants are more frequently establishing at higher elevations, and are becoming an increasing threat in these areas [16,23]. South Africa has relatively good records on the extent of alien plant invasions compared to many other countries, partly due to investment into large nation-wide initiatives such as the Working for Water programme (WfW) [24]. Yet, as in most areas of the world, this research has largely not been extended into higher-elevation mountain areas [25,26]. There is, however, growing evidence that alien plants are becoming more prevalent in the country's mountains. For example, it is estimated that over 170 alien plants have invaded the Maloti-Drakensberg [27], and a further 23 species have been identified as emerging invaders [28]. Within the same mountain range, Turner [15] found that the number of alien plants along the Sani Pass more than doubled during a decade (2007 to 2017). Road networks extending into montane areas are facilitating the establishment of alien plants beyond their elevational barriers and present sustained propagule pressure [26,29]. This threat has not been matched with appropriate expansion of mountain research and alien plant management interventions in South Africa.

Pauchard et al. [16] proposed a three-pronged global research agenda aimed at improving the understanding of plant invasions in mountain environments: (1) detection and analysis of invasion patterns at multiple scales; (2) experimental studies of invasion drivers; and (3) assessment of the impacts caused by alien species and their conservation implications. This paper addresses the first of these proposed research needs for South Africa—the documentation of patterns. Forming appropriate management programmes to protect mountain ecosystems will rely on improved understanding of the spread and occurrence of alien plants [17]. We assess the current status of alien plants in South African mountains by examining the sampling effort achieved in currently available species occurrence databases, the merit of existing alien plant sampling techniques, and assessing alien plant assemblages in terms of composition and abundances across mountain ranges and elevations (low- and higher-elevation).

## 2. Data and Methods

### 2.1. Delineating Mountain Ranges and Their Characteristics

Mountain areas were demarcated using a combination of Topographical Positional Index (an algorithm used to measure topographic slope positions and to automate landform classifications) and roughness surfaces [13,30]. These were used to produce six mountain area polygons in ArcMap 10.3 namely: the Western Great Escarpment (WGE); the Eastern Great Escarpment (EGE); the Southern Great Escarpment (SGE); Sub-tropical/Tropical Cuestas (TC); the Central Griqualand Mountains (CGM), and the Cape Fold Mountains (CFM) (Figure 1). From this, elevational gradient and vegetation types were determined across all six ranges using Schulze [31] for elevation and Rutherford et al. [32] for vegetation types.

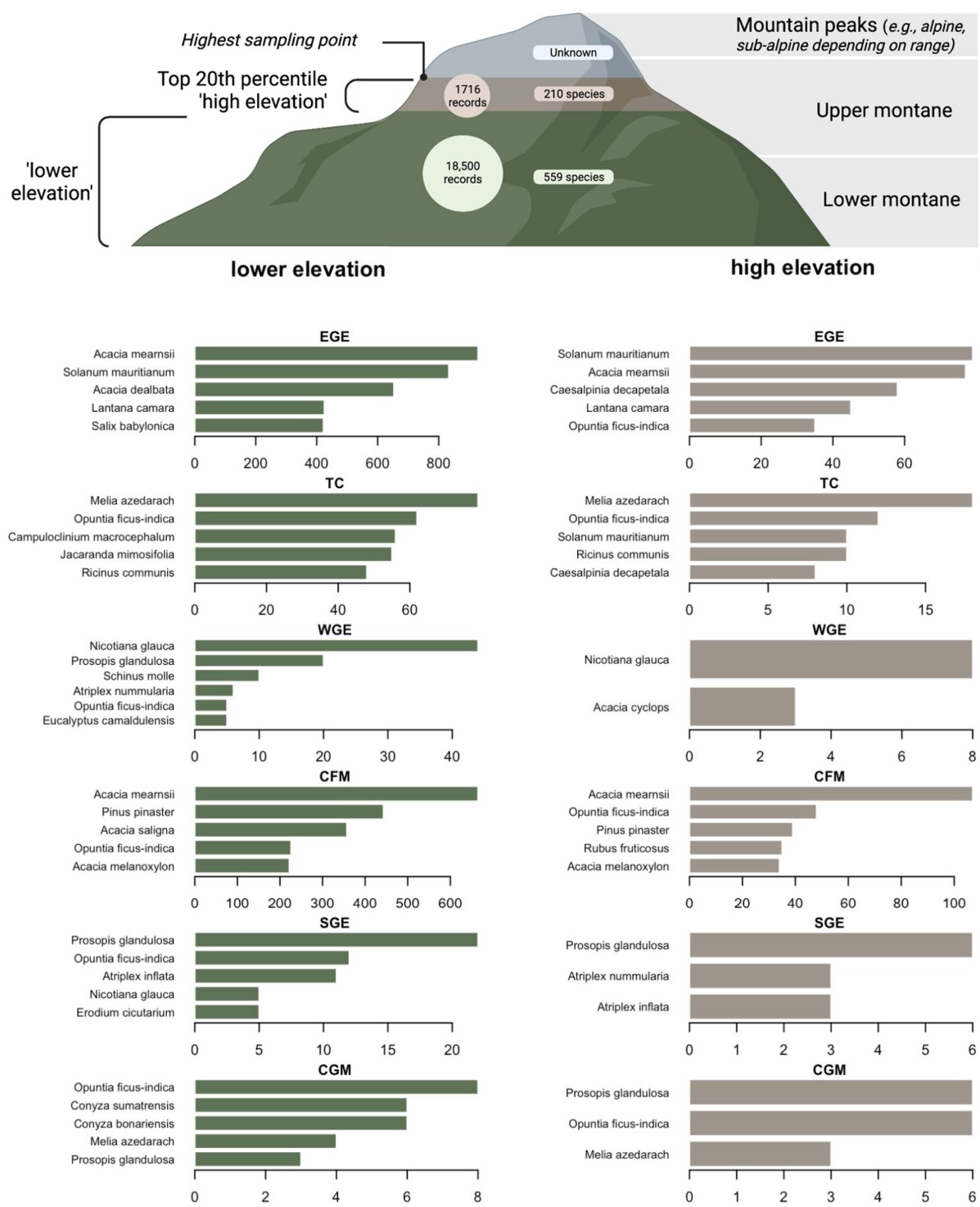

**Figure 1.** The top five most abundant alien plants recorded in each range in South Africa using SAPIA, iNaturalist and Great Escarpment Data (GED) databases. The high-elevation areas represent the upper 20th percentile elevational area in each range from the highest point recorded in the databases. The maximum elevation (highest peak) and high-elevation delineation for each range was—WGE: 1719 m, 1469 m; TC: 1868 m, 1528 m; EGE: 3446 m, 1651 m; SGE: 1784 m, 1451 m; CGM: 1605 m, 1355 m and CFM: 2064 m, 1667m. The lower elevational area reflect the entire mountain range below the designated high-elevation area. Figure created using BioRender (https://biorender.com/, accessed on 10 October 2021).

*2.2. Sampling Effort*

Sampling effort was assessed by performing a systematic literature review of all floristic surveys that include records of alien plants in South Africa's mountains. Alien plants were defined as species considered not native to the country and therefore "extralimitals" (species that are native to parts of South Africa but invasive in others) were not considered. Google Scholar [33], Scopus [34], ScienceDirect [35], Web of Science [36] and Taylor and Francis [37], databases were used. Furthermore, literature was obtained through searching bibliographies of papers and reports. There was no temporal constraint on journal publication date in the search. In September 2021, the following search terms items were included, 'floristic OR vegetation survey' AND 'South African mountains'; invasive OR weed OR alien AND plant species AND 'South African mountains'. Surveys were included if they stated GPS coordinates for all alien plant records to allow mountain area records to be extracted or were solely focused within the six mountain ranges of South Africa. After searching, the results were collated across databases and the title and abstract were used to select studies based on our criteria.

Three additional databases were included, as they are known to the authors to contain comprehensive records of alien plant occurrences—the Southern African Plant Invaders Atlas [38], iNaturalist [39] and the Great Escarpment Biodiversity Research Programme data mobilisation (hereafter referred to as GED) [40]. SAPIA is the best source of data on the distribution of alien plants in South Africa [24,41]. SAPIA was established to collate data on the distribution, abundance and habitat types of alien plants growing outside of cultivation in southern Africa [38]. iNaturalist is one of the largest online citizen science initiatives for naturalists, hosting about 50 million verifiable observations of 300,000 different species globally [42], and is an increasingly utilised resource for alien plant distribution in South Africa [43,44]. For this study, iNaturalist records were downloaded directly from the website and filtered for being verifiable, re-search grade, and alien plant species to South Africa (downloaded on 24/08/2021). The GED was a focused botanical survey (2005–2014) across two mountain ranges and is one of the most comprehensive publicly available mountain surveys [40].

Surveys that were carried out across more than one mountain range and contained GPS coordinates for all alien plant occurrences were analysed further. For these databases, alien plants recorded within the mountain layers were selected out in ArcMap10.3. All records of alien plants were included regardless of their invasive status (naturalised, casual etc.). The retention of all point observations may result in the inclusion of some duplicate records across the databases (duplicates were removed within each database). However, methods to avoid this duplication (i.e., removal of records of the same species within a grid-cell) would significantly reduce the dataset and retract from the aims of the paper including assessment of plant abundance.

The list of all species from each database was cleaned, updated and corrected; species names were checked for synonyms using the Plants of Southern Africa database (POSA) [45] and any revision of names were updated. When a species was not listed in POSA, the Plant List [46] was used to verify taxonomy. The native range for each species was determined using the Plants of the World Database [45] and was visualised using the 'maps' package in R [47]. For mapping, the ranges were generalised to the country-level (e.g., BrazilSouth to Brazil) and to update geopolitical boundaries (e.g., Czechoslovakia to Czech Republic and Slovakia). From this, 25 species were removed from the lists as they were noted to be alien; however, they have a native range within parts of South Africa (Table S1).

To assess sampling deficiencies on estimates of alien plant richness, we performed two analyses. Firstly, to assess whether sampling effort varies between mountain ranges we plotted the cumulative number of alien plant records and cumulative species richness over time for each of the six mountain ranges. Separate curves were plotted for each mountain range for records originating from the surveys to determine whether sampling effort variation between mountain ranges was linked to database (collector) bias. Secondly, we assessed whether sampling effort has varied between databases. A focused botanical

survey within a specific montane area (GED) was compared to two nation-wide surveys within the same geographical area. The cumulative number of alien plant records and cumulative species richness were plotted within the timeframe of the GED surveys. We then investigated whether there were taxonomic biases between databases by calculating the unique alien plant species within each family.

### 2.3. Species Composition

Alien plant assemblages were defined here according to Rouget et al. [48], who determined that invasive alien plants in South Africa cluster into distinct suites of species according to broad environmental conditions. Alien plant assemblages were assessed to determine how these species compositions varied between mountain range, databases and elevation (100 m intervals). To visualise how alien plant assemblages vary between mountain ranges, databases and elevation (100 m intervals), model-based unconstrained ordination was employed using the 'boral' R package [49]. A Bayesian hierarchical correlated response model was fit to the species-abundance matrix for the EGE and CFM ranges only. The remaining ranges had insufficient records for the analysis. The model uses latent variables to account for residual correlations between alien plant species. Thereafter, multivariate generalised linear models (MvGLM's) were used to test for the effect of mountain range, database and elevation on plant assemblages using the 'mvabund' package in R [50]. To do so, the 'manyglm' function was used to model the multivariate species abundances as the response variable, with mountain range (CFM, EGE), database (SAPIA, iNat) and elevational bands specified as categorical fixed effects. Likelihood-ratio tests (hereafter 'LRT') and pit-strap bootstrapping were used to compute P-values, using 999 bootstrap replicates, to assess the statistical significance of fixed effects.

Univariate GLM's were performed to determine which individual plant species accounted for any observed differences in plant composition between mountain ranges and elevation, using pit-strap bootstrapping to compute adjusted p-values corrected for multiple testing and correlations between species [50]. All GLM's were specified using a negative binomial distribution to account for mild overdispersion in preliminary models specified with a Poisson distribution and the strong mean-variance relationship present in the dataset. All statistical analyses were performed in R ver. 4.0.3. [51].

### 2.4. Species Abundance

The most abundant alien plants were determined for each range. To determine the species that have established in the highest elevational areas, each range was delineated into the highest-elevation areas (referred to as "high elevation") and then the adjacent lower-elevation areas (referred to as "lower elevation") (Figure 1). The highest elevational areas within mountain ranges generally have unique vegetative and climatic conditions that define the alpine and sub-alpine zones [52]. These distinct mountain zones are often differentiated by determining shifts in species composition and climatic conditions with elevation [16,53]. However, this could not be performed for this study for two reasons. Firstly, species composition assessments were not possible for all ranges due to a lack of alien plant records. Secondly, alpine and sub-alpine areas are only found within the EGE and CFM ranges respectively. Consequently, the highest areas of each range were categorised by demarcating the upper 20th percentile using the highest elevation point. When using the highest peaks for range as the upper limits, there were no alien plant records across all six ranges. The upper limit was therefore set as the highest elevational point that was taken during surveying for each range (see Figure 1). High-elevation areas were then defined as the upper 20th percentile from this high-elevation point and low-elevation areas were defined as the lower adjacent 80th percentile elevation. The five most abundant alien plants were then outlined for the highest and lower elevation areas for each range.

## 3. Results and Analyses

### 3.1. Delineating Mountain Ranges And Their Characteristics

The six major mountain ranges in South Africa were determined using the topographical positional index and roughness surfaces (Figure 2). Compared to mountains globally, South African mountains do not have particularly high absolute elevations, with a maximum elevation of 3446 m at the Mafadi peak in the EGE (the EGE reaches its highest point at 3482 m in Thabana Ntlenyana, Lesotho) (Figure 3). Overall, the EGE has the highest elevations in South Africa (with many points > 3000 m), while—of the other mountain ranges—only the CFM exceeds 2000 m (the highest point being Seweweekspoort Peak, 2325 m, in the Klein Swartberg); the WGE, SGE, TC and GCM are all < 2000 m (Figure 3). Elevational difference is more important for our purposes than absolute elevation, as the former provides the basis for determining surface area for invasion potential, and climatic partitioning and related potential invasion envelopes.

The WGE, EGE and SGE all form part of the same passive continental margin from the break-up of Gondwana, while the TC, CGM and CFM have diverse, ancient orogenies. Within these ranges, there is a strong effect of latitude on mountain vegetation—from temperate south to tropical north—and in terms of rainfall seasonality—from both south to north, and east to west [5,32,54]. A total of 443 vegetation types occur within these ranges according to Rutherford et al. [32] (Figure 3 and Table S2). The WGE—comprising the arid to hyper-arid Richtersveld and Namaqualand—has 36 vegetation types. The EGE—comprising the mesic grassy-dominated escarpment from the Sneeuberg to the Wolkberg—has 438 vegetation types. The SGE—comprising the arid Hantam–Roggeveld and Nuweveldberge—has 17 vegetation types. The TC—comprising mesic savanna-dominated hogsbacks in the far north—has 30 vegetation types. The CGM—comprising arid savanna—has seven vegetation types. The CFM—comprising fynbos-dominated folded mountains systems in a winter-rainfall region—has 139 vegetation types.

### 3.2. Sampling Effort

A total of 29 studies were found to include alien plant surveys within the mountains of South Africa (Table S3, references [55–75] are cited cited in the supplementary materials). Ten of these surveys focused only on recording alien plants. Most research has involved botanical surveys of specific mountain ranges with the aim of recording plant diversity and therefore focus has been on species richness rather than abundance. Most records of alien plants therefore came from lists produced during these botanical surveys. Only three databases include the GPS coordinates for each alien plant occurrence and covered more than one mountain range—SAPIA, iNaturalist and the GED (see Table 1 for comparison between databases).

A total of 570 alien plant species were recorded across all six mountain ranges, with 20,216 occurrence records across all three databases (SAPIA, iNaturalist and GED) (Table S4). Origins are from a wide range of countries, with China and Mexico being the source countries for the largest number of introduced taxa (Figure 4 and Table S4). The native ranges of 68 of the species were not included, as they were not represented in the POWO database. Most records were taken within the lower elevations of each range (18,500 total records) compared to the highest areas (1716 total records). The EGE and CFM had the greatest number of occurrence records, and also the highest density of sampling per area (Figure 2). The SGE and WGE had the fewest occurrence records per area (Figure 2).

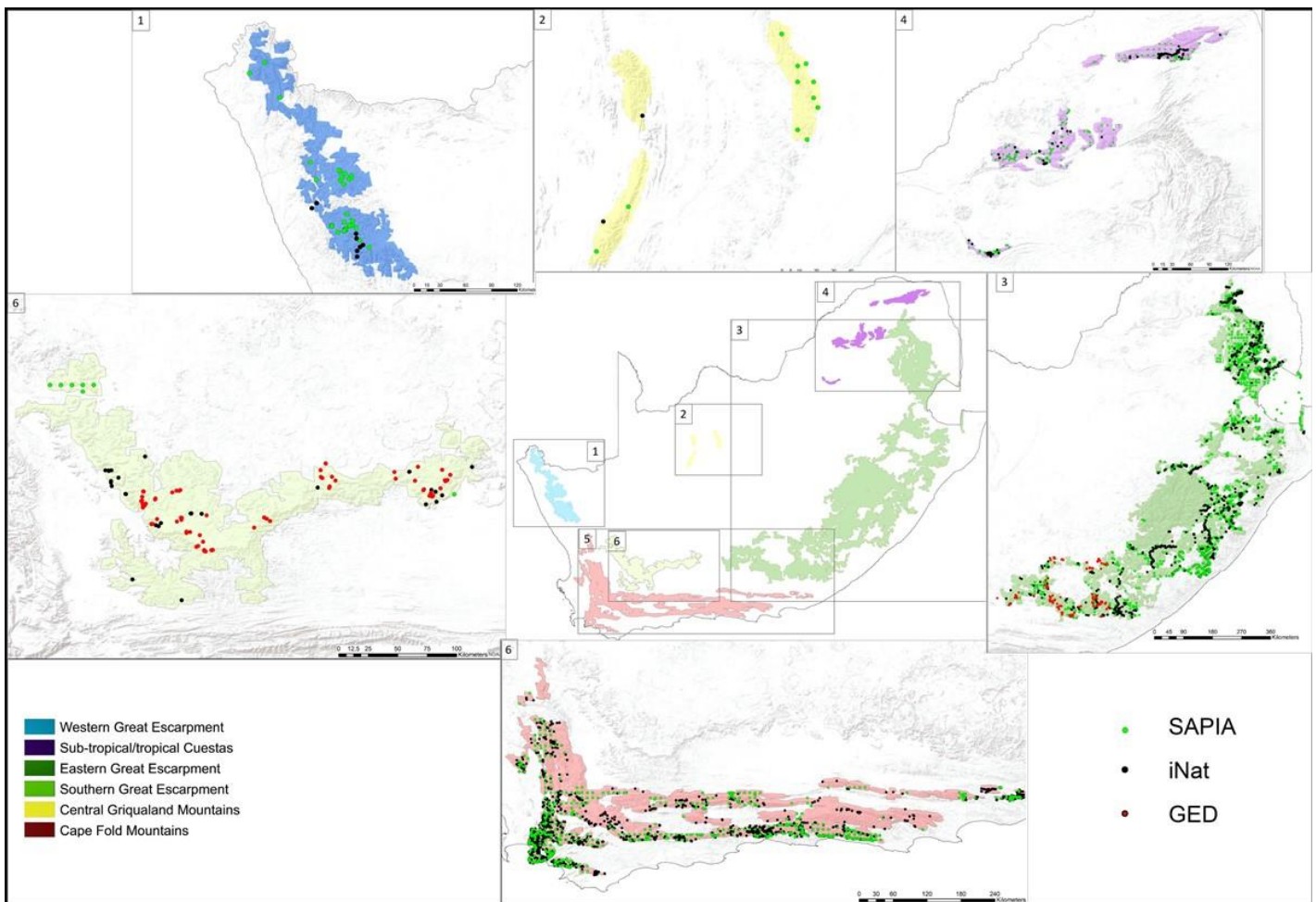

**Figure 2.** The six major mountain ranges in South Africa (adapted from Canavan et al. [13]), showing sampling effort across the six mountain ranges. The occurrence points for each alien plant record for all three databases is shown: SAPIA—the Southern African Plant Invaders Atlas, iNat—iNaturalist and the GED—the Great Escarpment Data.

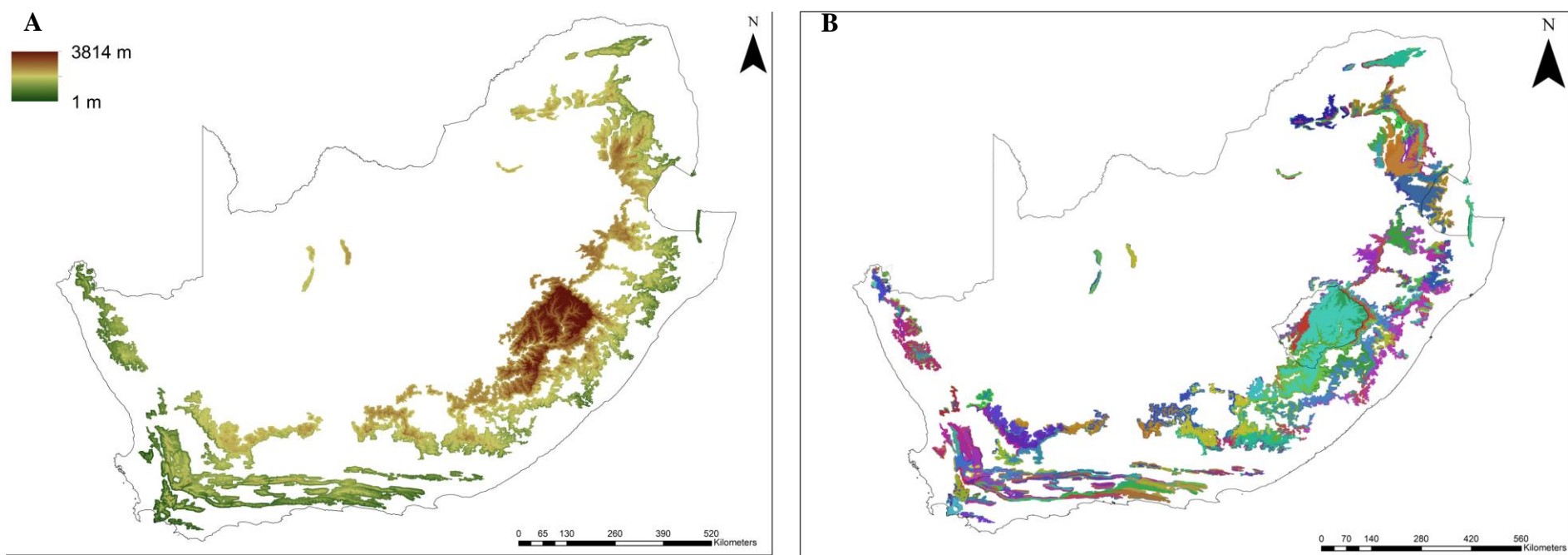

**Figure 3.** (**A**). The elevational gradient (metres) across the mountains of South Africa. (**B**). The vegetation types within South African mountains: EGE—438; TC—30; CGM—7; SGE—17; WGE—36, CFM—139 vegetation types according to Rutherford et al. [32] (see Table S2 and Figure S1 for vegetation type information).

**Table 1.** The three databases used to determine alien plant species occurrences in montane areas of South Africa.

| Database | Total Records in SA | Total Species Richness in SA | Total Alien Plant Records in Mountain Areas | Percent of Records in Mountain Areas | Total Species Richness in Mountains | Observers | Methodology | Level of Botanical Verification | Spatial Scale | Temporal Scale | Funding Dependent | Alien Specific | Source |
|---|---|---|---|---|---|---|---|---|---|---|---|---|---|
| SAPIA * | 80,226 | 969 | 18,278 | 26% | 588 | 710 | Nation-wide roadside survey, initially at the quarter degree scale, later using precise coordinates | High | National | 1979–2018 | Yes | Yes | Henderson and Wilson [55]; SAPIA Atlas [38] |
| iNaturalist * | 59,704 | 554 | 1472 | 3% | 239 | 9816 | Citizen science observations | Medium to high | Global | 2008–present (continuously updated) | No | No | Nugent [43] |
| GED | N/A | N/A | 466 | 100% | 160 | 3 | Botanist focused plant surveys | High | Regional | 2005−2014 | Yes | No | Barker [40] |

* SAPIA downloaded March 2018, iNaturalist downloaded 24 August 2021.

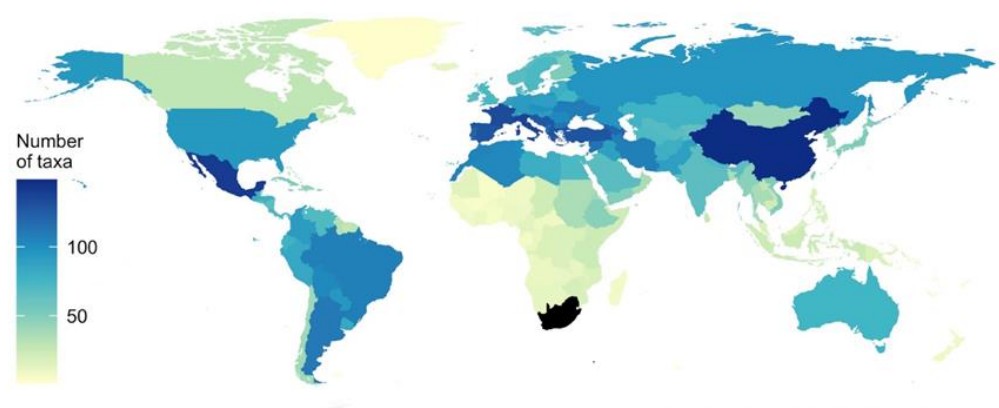

**Figure 4.** The native range distributions of the alien plants present within the six mountain ranges of South Africa (Table S3).

The nation-wide sampling effort for SAPIA and iNaturalist across all six mountain ranges revealed that all ranges have sampling deficiencies (Figure 5). The species accumulation curves for all mountain ranges did not reach an asymptote indicating that species richness of alien plants is not fully documented. Both the total number of records and alien plant richness are higher in SAPIA than iNaturalist. The occurrence data for the WGE, SGE and CGM is scarce and for both databases there has been little increase in numbers of records over time.

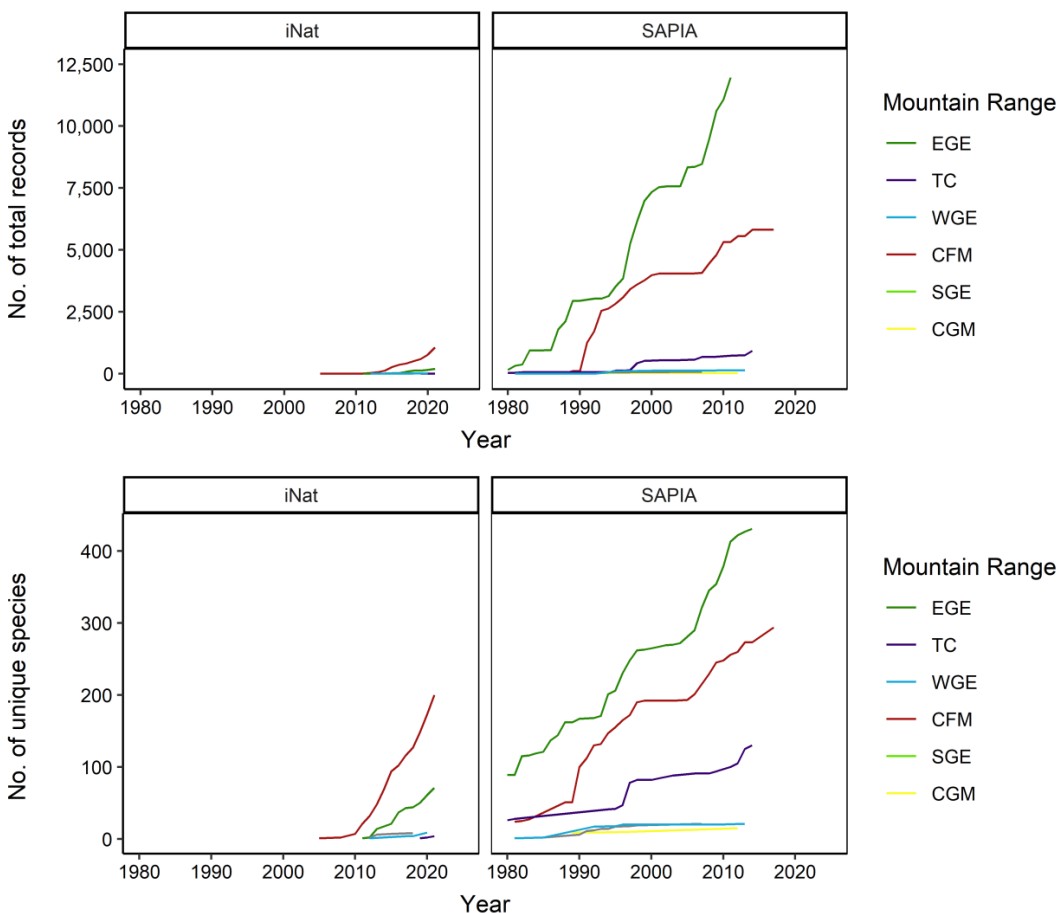

**Figure 5.** Species accumulation curves showing the number of records over time and the cumulative number of unique species recorded for iNaturalist and SAPIA databases for each mountain range across South Africa.

In assessing how sampling effort varied between the databases, the GED was found to have more effectively recorded the alien plants within the geographical area surveyed (Figure 6). A total of 160 alien species were recorded during the GED surveys over the nine-year sampling period, while SAPIA and iNaturalist combined recorded only 87 alien species with more than double the number of records over their full sampling periods (1994–2014, 2012–2014 for SAPIA and iNaturalist respectively). Species accumulation curves for the GED approached an asymptote, indicating that species richness was well documented (Figure 5). Within this surveyed area, a total of 41 plant families were recorded across all three databases, with 15 families being uniquely recorded in the GED (Table S3). The GED recorded 83 unique species (one of these being a new record for South Africa, *Sisymbrium runcinatum* (Brassicaceae)), with the Poaceae representing the most number of species (Figure 6). iNaturalist contributed only two unique species (Figure 6).

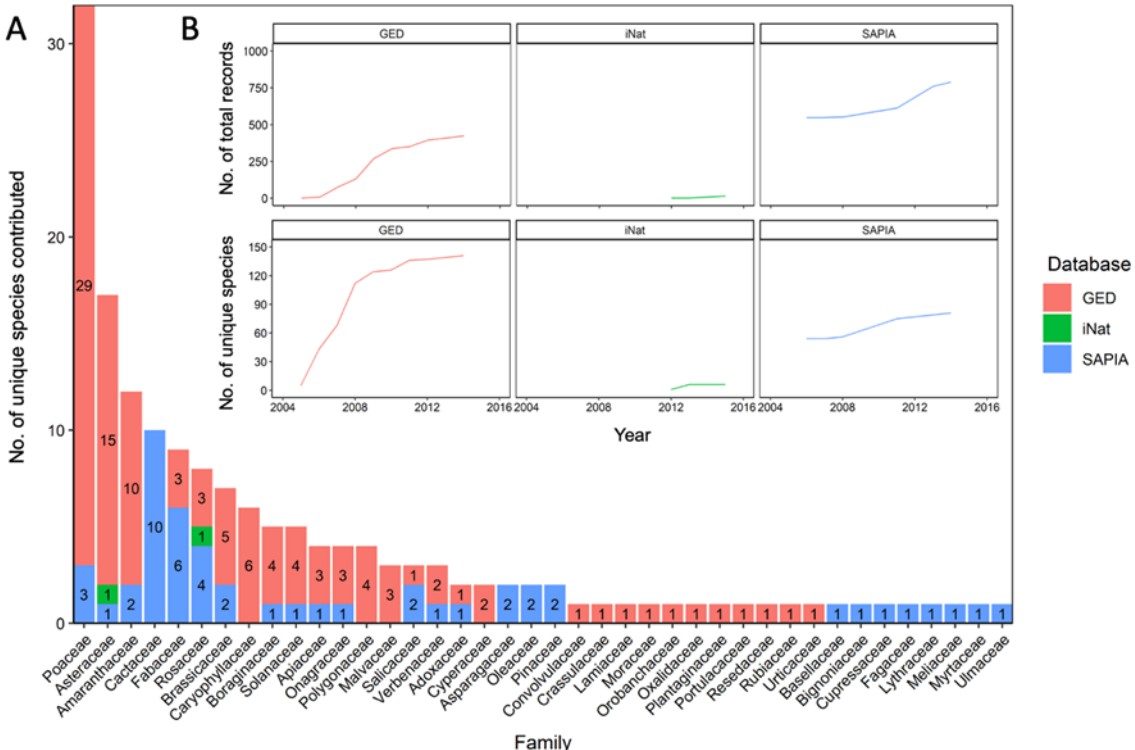

**Figure 6.** (**A**) The number of unique species recorded in each plant family across the three databases—Great Escarpment Data (GED), iNaturalist (iNat), South African Plant Invaders Atlas (SAPIA) (see Table S4 for all species recorded). Families are ordered by highest species richness from left to right. (**B**) Species accumulation curves showing the number of records over time and the cumulative number of unique species recorded for the GED, iNat and SAPIA. All records for each database are within the geographical range of the GED surveys (see Figure 3 for geographical area).

### 3.3. Species Composition

Species composition was only assessed for two ranges—the EGE and CFM. The remaining ranges had insufficient records to allow for meaningful species composition comparisons. There was significantly different species composition between the EGE and CFM mountain ranges ($X^2 = 2872$, d.f. = 1, $p < 0.01$), the two databases (iNaturalist and SAPIA) ($X^2 = 5045$, d.f. = 1, $p < 0.01$) and with elevation ($X^2 = 12,015$, d.f. = 17, $p < 0.01$). Univariate hypothesis tests determined which species were accounting for these differences (Table S4). For example, two species (*Hakea sericea* (Proteaceae) and *Pinus pinaster* (Pinaceae)) were found to drive the differences in elevation across the mountain ranges (Table S5); both species are largely restricted to lower elevations. There were 32 alien plants that were found to vary according to the mountain ranges where they were largely found to occur

within one range. For example, *Cotoneaster pannosus* (Rosaceae) was only found within the EGE. Sixty-nine plant species were found to vary according to the databases.

*3.4. Species Abundance within Each Range*

The higher elevation areas did not encompass the highest peaks of each mountain range due to a lack of records—and yet there were still very few records within these delineated areas (210 alien plant species, 1716 records) (Figure 1). The CFM had the greatest number of records in the higher elevation areas (above 1667 m) (Figure 1). Woody plant species made up the majority of the most abundant species in all ranges (Figure 1 and Table S6, references [76–83] are cited in the supplementary materials). The species found in the higher elevational areas were largely a subset of the most abundant species found in the lower elevational areas (70% of alien plants in high-elevations areas had abundant lower elevation populations) (Figure 1). For example, in the TC mountain range, *Melia azedarach* (Meliaceae) is the most abundant alien plant in both the lower and high-elevation areas. A majority of the most abundant species in both higher- and lower-elevation areas were frost tolerant and occur across a range of climatic zones (Table S6). All the species have capacity for long distance seed dispersal, with birds and water being the main vectors for spread (Table S6).

## 4. Discussion

Research on biological invasions in the mountainous areas of South Africa has been hampered by the lack of reliable baseline data for temporal studies, and sampling bias due to limitations of conducting surveys in challenging, inaccessible terrain [15]. This study has confirmed this and has highlighted major gaps in knowledge relating to alien plant distribution in South Africa's mountains. There has been an uneven sampling effort of alien plants across the country, and most surveying has been focused within the CFM and EGE. Higher elevation areas had considerably fewer alien plant records than adjacent lower elevation montane areas. All mountain ranges have sampling deficiencies, and the true extent of alien plant invasions is thus poorly known.

The SAPIA database contributed the most alien plant records, however these surveys are largely restricted to roadside observations. Such surveys capture only part of the spectrum of plant invasions in mountains and do not provide a true reflection of invasions across entire landscapes [25]. As such, data for the less accessible higher-elevation areas in the interior of South Africa are especially scarce. The surveying techniques of the GED and iNaturalist databases offer improved records as observers are often on foot and able to access entire landscapes, for example on hiking trails. At present, iNaturalist has contributed few occurrence records for most ranges, but this citizen science shows promise. For example, in the CFM where most iNaturalist observations were made, the number of occurrence records equated to more unique species compared to SAPIA (ratio of about 5:1 and 20:1 number of occurrence records to number of unique species for iNaturalist and SAPIA respectively). iNaturalist is a relatively new platform and has not yet been widely adopted. However, with increased public awareness, its use is likely to grow. Outreach campaigns for iNaturalist that are supported by easy-access communication channels have been found to be effective at obtaining new observations [84]. Overall, the GED surveying was most efficient at recording alien plants, supporting the need for focused botanical surveys and trained taxonomic experts. Yet, there has been a decline in field collection surveys in mountain areas in recent decades [85]. Continued support for such surveys will be highly valuable and will greatly improve the records for each range in South Africa.

Given the unique environmental conditions and vegetation types across different mountain zones, it was anticipated that the highest elevation areas would be invaded by different alien plant assemblages compared to lower adjacent areas. Evidence for this has been found in studies of high-elevation mountain areas in South Africa. For example, in the Maloti-Drakensberg in the EGE, surveys have found specific species becoming more abundant or even restricted to higher elevation areas such as *Cotoneaster*, *Pyracantha*, and *Rubus* spp. [13,28,29,52,86]. However, due to the lack of records found across all the

databases, the overall extent of invasion on South Africa's mountain peaks is unknown. Instead, the higher-elevation areas delineated here were below the mountain peaks, and the most abundant alien plants were found to be largely a subset of species that occur in the adjacent lower-elevation areas. This uniformity across elevations is in contrast to the native plant communities in these montane areas that are highly varied according to elevation and the environmental conditions across these landscapes (Figure 3); there are elevation-related vegetation differences particularly for CFM and EGE, as the relative elevation differences are the most ecologically significant in South African mountains: the CFM has a matrix of macchia/sclerophyllous-dominated vegetation (proteoid, ericoid, restioid, up to three to four m tall at maturity) that gives way to sub-alpine dwarfed forms of these >2000 m; similarly, the Maloti-Drakensberg in the EGE shows stratification, with $C_4$-grassland and evergreen forest mosaic <1800 m (montane zone), giving way to mixed $C_4$–$C_3$ sub-alpine grassland and sclerophyllous thickets (1800–2800 m), which in turn is replaced by alpine tussock grassland and ericoid shrubland >2800 m. The other mountain ranges show less discrete vegetation partitioning, but in general there is a trend of woody habitats prevalent at lower elevations transitioning to thinner woodland or pure sourveld grassland with increasing elevation. Most of the alien plants can however establish across a range of climatic conditions—and transgress these discrete native vegetation patterns—and have the ability for long-distance seed dispersal primarily by birds and water. It is likely that vertical dispersal of seeds is occurring whereby birds are carrying seeds into higher elevation areas and then water runoff moves the seeds downhill to establish populations in adjacent lowland areas. Vertical seed dispersal towards higher or lower altitudes is one of the critical processes for plant migration; for example, birds have contributed to the uphill seed dispersal of *Cerasus leveilleana* (Rosaceae) and *Prunus grayana* (Rosaceae) in two mountain ranges in Japan [87]. This pattern is consistent with global mountain studies whereby species that are reaching higher elevations are generally species from lowland invasions with broad ecological ranges and with the greatest capacity to adapt to novel conditions [14,21,22,88].

The unique characteristics within a mountain range will generally shape a distinctive assemblage of alien plants [89]. This was found to be the case for the EGE and CFM mountains, where the species compositions of alien plants were found to be significantly different between the ranges. This variance is likely a reflection of their distinct environmental conditions, including climate—which has been found to have the greatest impact on the composition of invasive alien plants in South Africa [48]. The CFM and EGE occur in different climatic and vegetation types, being predominantly within the Fynbos and Grassland biomes, respectively [32]. In addition, their histories of human influence have contributed to the extent and types of alien plants present. The CFM have had much greater exposure to anthropogenic disturbance (international trading through the Cape trade route since 1652) [90], whereas most areas in the EGE have had relatively recent exposure (from the late 1700s in the south and from c. 1850 in the north) [91]. While species composition assessments were not possible for the remaining mountain ranges, the variance determined in their most abundant species indicates the likelihood of distinct alien plant assemblages.

The most abundant species in all ranges were largely woody species, showing a pattern of woody plant encroachment (WPE). Woody plant encroachment is becoming more prevalent in the region. For example, over the past three decades, 7.5 million $km^2$ (55%) of non-forest biomes in sub-Saharan Africa have had significant net gains in woody plant cover [92]. While there is clear evidence of WPE in South Africa's mountains, it is also important to recognise that surveys, particularly SAPIA, have been biased towards this group [38]. The majority of alien plants in alpine areas globally have been found to be herbaceous species, as these areas generally support low-growing shrubs and grasses [11,52]. Alexander et al. [17] found that most alpine alien plants were in the families Poaceae, Asteraceae, Caryophyllaceae, Fabaceae, and Brassicaceae, in order of abundance. Floristic surveys in the Maloti-Drakensberg have supported this, with Poaceae and Asteraceae contributing the most invaders [27]. It is likely that herbaceous and (particularly) grass

species [93] have been undersampled in South Africa's mountains. This was evident through the inclusion of the focused botanical GED surveys where there were considerably more unique species and families recorded, most of which were herbaceous and grass species. The experience of the botanists conducting these surveys allowed for the detection of plant families that are often difficult to identify taxonomically. For example, the invasive alien grass, *Nassella trichotoma* (Poaceae), was only recorded in the GED survey. This species is considered a morphologically cryptic species in South Africa as it is very similar to certain native grasses, which means that populations often go unnoticed [94]; such "cryptic" alien plants are not easily amenable to citizen science efforts (such as iNaturalist) and require specialist searches (e.g., Sylvester et al. [93]).

The bias towards the recording of woody plants is also a reflection of their disproportionate impacts in these ecosystems [11]. Although the spatial combinations of vegetation communities in South African mountains is complex [32], most montane areas in South Africa (particularly the higher-elevation areas) are open, tree sparse habitats structurally dominated by graminoids (grasses, sedges, and restios) in the wetter mountains, and shrubs in the drier mountains; only the TC and CGM have spatially dominant natural woody elements at higher elevations (e.g., *Protea* and *Faurea* woodland communities in the TC) [52,95]. When large alien invasive trees and shrubs plants establish in these ecosystems, there is often a major shift in their proportion of the plant biomass compared to native species. According to the biomass-ratio hypothesis, ecosystem properties are driven by the traits of dominant species in the community, and these are generally those with the greatest biomass [96]. Woody plant encroachment therefore has the ability to transform these ecosystems [52,95]. One of the most concerning consequences of this is a change to the hydrology of the water sources that can lead to greatly reduced water availability [97].

At present, management of alien plants in mountains is also primarily focused on woody species, with particular investment on species of the *Acacia*, *Hakea* and *Pinus* [88,98]. Unfortunately, because of the costs of control, and the logistical challenges in managing the biomass, especially relating to fire—the control of invasive woody species is more challenging compared to other invasive alien plants [99]. Additionally, in mountain regions, alien species, including trees, often grow on steep slopes and in dangerous terrain where conventional control methods are difficult, expensive, and carry high risk to personal safety [100]. Management of alien plants in South Africa is currently largely coordinated through the WfW programme, which works in partnership with local communities [101]. For species growing in upper catchments and in rugged terrain, the programme has specifically developed "high-altitude teams" [102]. These teams only reach a fraction of the areas requiring management. Due to such difficulties, WfW has also supported the development of The Northern Temperate Weeds programme, which aims to use biological control to target some of the problematic alien plants in mountain regions [100]. However, this programme will only be able to offer solutions for a select few alien species. Effective management thus remains a major challenge in mountain regions in South Africa, which is only exacerbated by the lack of concise data. Until suitable data are obtained, we suggest an area-based management approach whereby management efforts are prioritised in areas where the greatest impacts occur now or are projected to occur in the future [103].

## 5. Conclusions

Mountains are one of the few ecosystem types where proactive management of alien plant species may still be possible [88]. However, in South Africa, convincing funding bodies to invest in alien plant management will require comprehensive knowledge of the identity and distribution of the most problematic species. The outlook for improved detection and analysis of alien plant patterns is promising, with expanding local research and greater opportunities for local actors to collaborate with global networks. The Afromontane Research Unit of the University of the Free State is facilitating the expansion of this work through partnerships with international groups such as the Mountain Invasion Research Network (MIREN) [104]. Based on the findings of this study, it is recommended that these

initiatives expand an alien plant inventorying, targeting understudied ranges (e.g., SGE and CGM) and employing focused botanical surveys to allow for fine scale sampling with GPS-recorded localities. In addition, the use of iNaturalist presents a cost-effective way to further enhance alien plant records and outreach campaigns that highlight mountain areas are likely to be beneficial. This would provide the means to begin establishing a comprehensive national mountain database on alien plants to guide strategic planning at regional and national levels.

**Supplementary Materials:** The following are available online at https://www.mdpi.com/article/10.3390/land10121393/s1, Table S1: Plant species removed from lists due to erroneous inclusion in databases as alien species to South Africa; Table S2: The vegetation types across the six mountain ranges in South Africa according to Rutherford et al. [32]. Table S3: Mountain surveys that have been conducted across the six mountain ranges in South Africa; Table S4: The alien plant species found across all databases—SAPIA, iNaturalist and the Great Escarpment Data (GED); Table S5: Univariate hypothesis tests for alien plant differences between databases (SAPIA and iNaturalist), mountain ranges (Eastern Great Escarpment and Cape Fold Mountains) and elevation. Highlighted values indicate significant differences ($p < 0.01$); Table S6: The most abundant alien plants found across all six mountain ranges in South Africa (see Figure 1). Figure S1: Legend of vegetation types across the six mountain ranges in South Africa (see Figure 3). Vegetation types not included in the map had very restricted geographical ranges (see full list of vegetation types in Table S2).

**Author Contributions:** All authors conceived the idea of the study. K.C. led the writing with input from G.F.S., S.C., O.G., G.D.M., V.R.C. and D.M.R.; G.F.S. performed the statistical analyses and plotted the results; S.C. led data curation and visualisation; O.G. performed the literature review; G.D.M. produced the maps and supervised the project. All authors have read and agreed to the published version of the manuscript.

**Funding:** K.C., G.F.S. and G.D.M. acknowledge the funding from the South African Working for Water (WfW) programme of the Department of Forestry, Fisheries and the Environment: Natural Resource Management Programmes (DFFE: NRMP). Funding was also provided by the South African Research Chairs Initiative of the Department of Science and Technology and the National Research Foundation (N.R.F.) of South Africa. V.R.C. and O.G. thank the Afromontane Research Unit through the University of the Free State and the Oppenheimer Memorial Trust for funds for post-doctoral support (O.G.) and running costs (V.R.C.). D.M.R. acknowledges support from the DST-NRF Centre of Excellence for Invasion Biology. Mervyn Lotter is acknowledged for creating the 'Mountain Areas— southern Africa' shapefile. Any opinions, finding, conclusions or recommendations expressed in this material are those of the authors and the NRF does not accept any liability in this regard.

**Institutional Review Board Statement:** Not applicable.

**Informed Consent Statement:** Not applicable.

**Data Availability Statement:** The data presented in this study are available in Supplementary Tables S2, S4 and S5.

**Conflicts of Interest:** The authors declare that they have no conflict of interest.

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
