# Peer review of "The Alien Plants That Threaten South Africa’s Mountain Ecosystems"

_land, doi:10.3390/land10121393_

Round 1

Reviewer 1 Report

This study on alien plant species and their distribution is interesting. I think the manuscript represents well a relevant problem, namely the collection of data on alien occurrences in high conservation value regions such as mountains. Giving evidence of the most invaded areas and the most invasive species to be monitored nationwide. However, I found some issues that I would ask you to consider for improving the article or for future research.

I have three general questions/observations about the authors’ research:

  1. Why did you not search on other databases (e.g., GBIF https://www.gbif.org/; GISD http://www.iucngisd.org/gisd/) other than those three mentioned? Especially for those areas (SGE and WGM) where you did not find enough data.
  2. How did you handle the point observations? Were they spatially spaced, or could there be several records for the same species at the same location? Usually when working with point observations it is common to eliminate those that are closer than a certain threshold to avoid redundancy.
  3. I think a table or a description of the most common native plant communities along the elevation gradient might help visualize and better understand the invasion pattern. In the abstract is stated: “This study assesses the status of alien plants in South African mountains by determining sampling effort, species compositions and abundances across the six ranges in lower-and higher-elevation areas” but except for Figure 4 and 5 the information about alien species composition is missing (Table S3 is not present in the supplementary materials as well as Table S4). A couple of lines about the native communities invaded by the alien species showed in Figure 5 would be helpful. Often the same alien species is the most abundant at both altitudes, and I expect the native communities to be different instead.

Further minor corrections/suggestions:

I found several “dashed” words (e.g., ap-propriate, tradi-tional), please check throughout the manuscript and correct them.

References do not follow the journal standards: in the instructions for authors section is stated that “References must be numbered in order of appearance in the text (including table captions and figure legends) and listed individually at the end of the manuscript”.

I also noticed that in the Supplementary there are references not reported in the reference list. “Citations and References in Supplementary files are permitted provided that they also appear in the main text and in the reference list”. Please update the references according to the journal instructions.

METHODS

Figure 1: “The occurrence points for each alien plant record for all three databases is shown”. I would suggest to the authors improving the image resolution and/or changing the background colors. It is impossible to distinguish between different shapes and the points are also difficult to sea in Cape Fold Mountains.

RESULTS

L 243 Table S3 does not exist. Please add it to the Supplementary files.

LL 291-292 Species’ names should be written in italic and without the author’s name.

L 295 View previous comment.

LL 305-307 “The species found in the higher elevational areas were largely a subset of the most abundant species found in the lower elevational areas (70% of alien plants in high-elevations areas had abundant lower elevation populations)”. This is very interesting. As already said, it would be helpful to know how the native communities are structured at different elevations in order to explore if the aliens’ success is due only to their dispersal ability and their tolerance to different climate ranges or also to their functional traits that are able to outcompete native species.

L 308 Species’ names should be written in italic and without the author’s name.

LL 311-312 Is it possible that this could create a sort of vertical dispersion? Seeds are transported by birds at higher or lower elevations and at the same time seeds from higher elevation plants can go downward through runoff. If yes, I would comment it in the discussion part relatively to the fact that the species found in the higher elevational areas were largely a subset of the most abundant species found in the lower elevational areas.

L 317 Table S4 does not exist. Please add it to the Supplementary files.

L 341 Hasn’t. Please write “has not”.

L 375 “The CFM and EGE occur in different climatic and vegetation types” This sentence made me curious to know more about native vegetation in the study area. See the third observation at the beginning of this review.

L 403 Species’ names should be written in italic and without the author’s name.

L 426 Genus’ names should be written in italic.

Reviewer 2 Report

Invasive alien plant species are recognized as one of the most important threats to biodiversity worldwide. Therefore, the international community is taking measures to prevent and limit the spread of invasive species. In this regard, the submitted manuscript entitled “The alien plants that threaten South Africa’s mountain ecosystems” is of interest to a wide readership. The importance of such studies, especially in less studied regions, is undeniable. The scientific questions raised in this study are well defined and original, at least in terms of the geographical scope of South Africa. The proposed information does not provide original new data on invasive species in the study region because it refers to available sources of information, but presents this information in summary and synthesized form. As a person who have never been in South Africa’s mountains, I consider this manuscript informative. The results are significant because they outline the gaps in knowledge. This is the main message of the study. However, the manuscript needs some improvement.

General comments

"Alien plant communities" is used several times in the text. Since the common understanding of plant communities includes a variety of species that live together and share the ecological conditions of the habitat, I think it is necessary the authors to clarify how they understand this concept. Is “unique species composition” determined by the presence of alien species only (L.63) or means composition of single species individuals?

In Section 2.1. the altitudinal gradient is very poorly characterized. It would be useful to indicate the average and maximum heights of the six mountain regions. Despite their climatic differences, as well as differences in terms of vegetation, this will guide readers who do not know the orography of the mountains. Additionally the Figure 1 as presented now does not allow to identify the records from the three databases. I suggest only one type of points to remain on the figure. Section 2.4 should be edited to make it more understandable. I had to look carefully at Figure 5 to find my way around. I suggest for this figure the division of the mountains into an altitudinal gradient to be accompanied by a meters above sea level.

In the Results section, the abbreviation WGM was used, which is not described (L. 250). Figure 1 is essentially a result and should be part of the results. Figure S1 is not informative. The authors also indicate this in the text (L.298). I suggest to omit this figure. Table S6 contains various data for the most numerous species, but in order to be convincing that woody plants are the most numerous, there must be a particular number of records or leave only Figure 5 which illustrates very well the five most abundant species. On Line 303 the ‘Figure 5 ‘should be corrected to ‘Figure 1’.

Technical notes

Please review the entire text and remove unnecessary hyphenations, such as lines 57, 58, 59, 402, 412, 461, and many more. Tables S3 and S4 in the supplementary material are missing, they have only titles. The cited Figure 7 is also missing. Please add a title to the x-axis of the graphs in Figure 5; presented in this way, it is not clear what the abundance measure is. I suggest replacing “unique species” with “discrete” or “single species”. Line 364 - the cited authors have one redundant bracket.
